# Effects of Coolant and Working Temperature on the Cavitation in an Aeronautic Cooling Pump with High Rotation Speed

Ao Wu [1], Ruijie Zhao [1,*], Fei Wang [2], Desheng Zhang [1] and Xikun Wang [1]

[1] Research Center of Fluid Machinery Engineering and Technology, Jiangsu University, Zhenjiang 212013, China
[2] Aviation Key Laboratory of Science and Technology on Aero-Electromechanical System Integration, Nanjing 211106, China
* Correspondence: rjzhao@ujs.edu.cn

**Abstract:** The centrifugal pump with high rotation speed is the key component in the cooling system of an aircraft. Because of the high rotation speed, the impeller inlet is very prone to cavitation. Two impellers with different types of blades (cylindrical and splitter) are designed, and the numerical models of the pumps are built. The authenticity of the numerical models is validated with the corresponding experiments in terms of both the hydraulic and cavitation characteristics. Then, the effects of different coolants and working temperatures on the hydraulic and cavitation performances of the prototype models are studied based on the numerical simulations. The results show that the head and efficiency of the pump for conveying water are higher than those for conveying ethylene glycol (EG) aqueous solution and propylene glycol (PG) aqueous solution (EGaq and PGaq are defined to represent the EG aqueous solution and the PG aqueous solution, respectively). The hydraulic performance in the EGaq is slightly better than that in the PGaq. The cavitation performance of water is far less than that of the EGaq and PGaq under high working temperature. The volume of cavitation in EGaq is smaller than that in PGaq, and the volume of cavitation in the splitter blades is slightly smaller than that in the cylindrical blades. It is suggested that EGaq be used as the first option. The splitter blades can improve the cavitation performance somehow while the improvement by using the splitter blades is very limited at high rotation speeds, and the design of the short blades should be careful in order to obtain a smooth internal flow field.

**Keywords:** aviation liquid cooling pump; pump performance; cavitation; numerical simulation; organic coolant





## 1. Introduction

As the power of onboard electronic devices is continuously increased, the liquid cooling pump with high performance and reliability is critical to maintain the working environment for such devices. A small pump diameter and high rotation speed are the main characteristics of the centrifugal pump used in aircraft cooling systems because the aerial environment limits the size and weight of the cooling pump, and a high rotation speed can provide sufficient pressure head to circuit coolant in the cooling system [1]. Such pumps are also termed as the low-specific-speed pump, in which the flow passages of the impeller are long and narrow, and the fluid velocity at pump inlet is high [2,3]. Therefore, the flow is prone to cavitation, which will change the pattern of internal flow and severely deteriorate the pump performance.

Cavitation in pumps has been intensively studied during the last several decades. Rayleigh [4] took spherical symmetric cavitation as research object and put forward the famous Rayleigh equation. Plesset et al. [5,6] considered that the effects of gas, fluid viscosity and surface tension contained in the cavitation improved the cavitation dynamics theory and formulated the Rayleigh-Plesset equation. Brennen et al. [7] compared and discussed the dynamic transfer functions of two cavitating inducers with the same geometry but

different dimensions from the experiments and further confirmed the validation of the theoretical model. Katz [8] studied the phenomenon of cavitation on four axisymmetric bodies whose boundary layers underwent a laminar separation and subsequent turbulent reattachment. The conditions for cavitation inception and desinence were determined by holography and the Schlieren flow visualization technique, and several holograms were recorded just prior to and at the onset of cavitation. Tanaka et al. [9] experimentally investigated the transient behavior of a cavitating centrifugal pump and found that the oscillating cavitation during the pump start-up and the separation of water column during the sudden pump shutdown are the main reasons for the fluctuations in the pump pressure and flow rate. Huang [10] evaluated Four cavitation models, including the Kubota model, Singhal model, Merkle model and Kunz model in the present study, which provides a theoretical and technical basis for the research of cavitation numerical simulation. Cheng et al. [11] summarized the research progress of cavitation, including cavitation characteristics, numerical methods, and the impact of cavitation on the flow field and proposed some frontier topics, which are of great significance for promoting cavitation research.

Computational Fluid Dynamics (CFD) technique has become popular during the last two decades for research on fluid problems in pumps. Besides accurate prediction of pressure and fluid velocity in a pump, turbulence should also be precisely predicted since occurrence and evolution of cavitation are strongly correlated with the local turbulence [12]. Much effort has been devoted to choosing a proper turbulence model for modeling the flow in pump [13–15]. Sun et al. [16] used the CFD method to simulate the internal flow in a pump used as turbine under different working conditions, and the comparison between the simulation result and the experiment demonstrated that the Detached Eddy Simulation (DES) model performed better than all models in the family of two-equation models. Feng et al. [17] performed both the steady-state and transient simulations of a centrifugal pump at design and off-design conditions by employing different turbulence models, and the results indicated that the choice of turbulence model has little effect on the prediction of pump head and energy efficiency. The standard $k$-$\varepsilon$ model can obtain better results for the fluid velocity while the shear stress transport (SST) $k$-$\omega$ model was superior in the prediction of turbulent parameters. Zhou et al. [18] simulated a double-suction centrifugal pump with a specific-speed of 120 using both the S-A DES (one-equation DES) model and the SST $k$-$\omega$ model. The results show that both models can predict accurately the required power and pressure pulsation at different flow rates, and the performance of the former one is slightly better. Zhang et al. [19] employed the Delayed Detached Eddy Simulation (DDES) model to simulate the stalling effect in a centrifugal pump with low speed, and the simulation results reproduced well the transient flow behaviors and structures, especially for the jet wake at the outlet of the impeller blade.

The numerical simulation of cavitation in low-specific-speed pumps has been also received much attention by researchers. Wang et al. [20] modified turbulence models, such as the Re-Normalization Group (RNG) $k$-$\varepsilon$, SST $k$-$\omega$, and Filter-based model (FBM) through the density correction, and used them for the prediction of cavitation performance of a low-specific-speed centrifugal pump. The results predicted by the modified turbulence models were in better agreement with the experiments. Phillip et al. [21] used the commercial CFD software CFX to simulate the cavitating flow in a low-specific-speed centrifugal pump under different working conditions and surface roughness. The simulation result was in good agreement with the corresponding experiment in the non-cavitating state. In addition, the inception of cavitation at the diaphragm tongue can be predicted more accurately by using the low-Reynolds-number methods. Gao et al. [22] analyzed the relationships between cavitation and vibration characteristics of a low-specific-speed centrifugal pump based on the experimental and simulation results, specifically focusing on the relationships between 10~500Hz low-frequency vibrations and cavitation evolution.

The effect of the impeller's structure on the behavior of cavitation in low-specific-speed pumps was also studied in some works. It was found that the cavitation can be effectively suppressed when the traditional impeller blade is replaced by a combination

of long and short blades, or splitter blades [23]. Zhang et al. [19] numerically studied the effect of the curvature of short blades on the pressure fluctuation in a low-specific-speed centrifugal pump under different flow rates. It was found that if the outlet angle of the short blades was set as 12° to the pressure side of the blades, the streamlines in the flow channels were more in line with the blade shape, and both the overall pressure fluctuation at the pump outlet and the turbulence intensity were decreased. Hu et al. [24] carried out numerical simulation and hydraulic test for a low-specific-speed centrifugal pump under three attacking angles to study the effect of the inlet attacking angle of blade on the development of cavitation in the impeller, as well as the evolution of cavitating flow in the flow channels. Results show that as the cavitation number decreases, the cavity first occurs at the suction side of the leading edge of the blade and then expands to the outlet of impeller rapidly along the blade. The load of the blade near the tongue is heavier than that of other blades whether cavitation occurs or not.

From the above literature review, it is clear that the two-equation turbulence models are usually employed in modeling cavitation in pumps, probably because their predictions of the inception and evolution of cavitation are of reasonable accuracy. The more advanced turbulence models, such as DES, DDES, and LES, can predict more details of the flow structure in pumps while the required computational resources are also significantly increased. As for the cavitation in the aeronautic cooling pump, the material properties of the coolant are very different from those of pure water, and they are also more sensitive to the temperature [25,26]. The effects of cooling media and working temperature on the cavitation behavior in such pumps have yet to be revealed.

In this work, a pump model with two types of impeller's blades is firstly designed for an aircraft cooling system. The numerical model of the pump based on CFD is built and introduced in Section 2. The simulation procedures are validated with the corresponding experiments in terms of both the hydraulic and cavitation performances in Section 3. The effects of different coolants and working temperatures on the pattern of hydraulic performance and cavitation in the pumps are studied based on the simulation results in Section 4. The conclusions are summarized in the end.

## 2. Numerical Model

### 2.1. Physical Model

In this paper, a pump model with two types of impeller blades is designed based on a set of the design parameters of an aeronautic cooling pump. The two impellers are of cylindrical and splitter blades. To design a low-specific-speed pump, Yuan et al. [27] proposed enlarging the design flow rate and the specific-speed in order to improve the pump efficiency at the design point. The enlarged design flow rate and specific-speed are determined by Equations (1) and (2):

$$Q' = K_1^2 Q \tag{1}$$

$$n_s' = K_2 n_s \tag{2}$$

where $Q$ and $Q'$ denote the original design and modified design flow rates, respectively. $n_s$ and $n_s'$ indicated the original and modified design specific-speed, respectively. $K_1 = 1.2$, $K_2 = 1.17$ are the magnification index.

The design parameters of the prototype pump studied in this paper are presented in Table 1. The original dynamic specific-speed of the pump $n_s'$ is calculated as:

$$n_s = \frac{3.65 n \sqrt{Q}}{H^{3/4}} \tag{3}$$

where H is the pressure head in the unit of m and n the rotation speed in the unit of r/min. According to Equations (1) and (2), the enlarged design flow rate is $Q_d' = 15.84 \frac{\text{m}^3}{\text{h}}$, and the enlarged specific-speed is $n_s' = 73.04$. Then, a new design pressure head is calculated

according to Equation (3). The modified design parameters are also shown in Table 1. The main parameters of the model pump are shown in Table 2. Two blade shapes are designed based on the modified design parameters, and the water bodies of the corresponding CFD model are shown in Figure 1a–c. It is noted that an extra pipe with a length of 8D is modeled in front of the pump inlet, and the outlet pipe is extended to 10D to ensure that the turbulent flow is fully developed.

**Table 1.** Design parameters of the aeronautic cooling pump.

|  | Flow Rate $Q_d$ (m³/h) | Pump Head $H$ (m) | Speed n (r/min) |
|---|---|---|---|
| Original design parameters | 13.2 | 132 | 11,000 |
| Modified design parameters | 15.84 | 120.91 | 11,000 |

**Table 2.** The model pump main parameters.

| Parameters | Value |
|---|---|
| Rated speed $n_d$ | 11,000 r/min |
| Specific speed $n_s$ | 62 |
| Impeller inlet diameter $D_1$ | 32 mm |
| Impeller outlet diameter $D_2$ | 94 mm |
| Impeller Outlet Width $b_2$ | 4 mm |
| Volute outlet diameter $D_4$ | 20 mm |

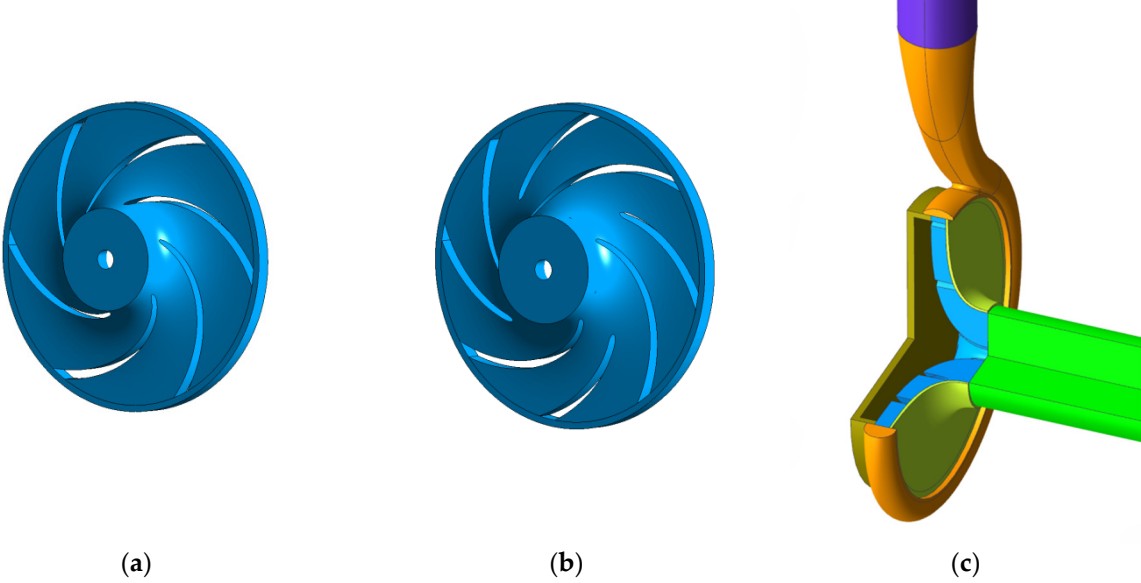

| (**a**) | (**b**) | (**c**) |

**Figure 1.** Water bodies of the two impellers and volute of the CFD pump model: (**a**) cylindrical blade impeller; (**b**) splitter blade impeller; (**c**) pump model.

## 2.2. Governing Equations

The vapor volume fraction transport equation is added in the homogeneous multiphase flow model to simulate the formation and evolution of cavitation. The conservation equations of continuity and momentum of a fluid can be written in the following formula:

$$\frac{\partial \rho_m}{\partial t} + \frac{\partial (\rho_m u_i)}{\partial x_i} = 0 \tag{4}$$

$$\frac{\partial(\rho_m u_i)}{\partial t} + \frac{\partial(\rho_m u_i u_j)}{\partial x_j} = -\frac{\partial p}{\partial x_i} + \frac{\partial}{\partial x_i}\left[(\mu_m + \mu_t)\left(\frac{\partial u_i}{\partial x_j} + \frac{\partial u_j}{\partial x_i} - \frac{2}{3}\frac{\partial u_k}{\partial x_k}\delta_{ij}\right)\right] \tag{5}$$

where $\rho_m$ and $\mu_m$ represent mixing density and hybrid viscosity, respectively, and they are expressed as

$$\rho_m = \rho_l \alpha_l + \rho_v \alpha_v \tag{6}$$

$$\mu_m = \mu_l \alpha_l + \mu_v \alpha_v \tag{7}$$

The subscripts *l*, *v*, and *m* represent liquid phase, gas phase, and mixed phase, respectively; *p*, *u*, and $\mu_t$ stand for pressure, velocity, and turbulent viscosity, respectively; *i*, *j*, and *k*, respectively, represent the three directions of the coordinate system, without considering the gravitational action term. Due to the little effect of the gravity in the pump of high pressure head, the gravitational force is usually ignored in the simulation [28,29].

The SST *k-ω* turbulence model is used in the numerical model [30,31]. It is a combination of the *k-ω* and *k-ε* models. It takes full advantage of the superior performance of the *k-ω* model in predicting the near-wall region and the *k-ε* model in predicting the far-field free flow. The performance of the inverse pressure gradient boundary layer model is significantly improved. The SST *k-ω* turbulence model is derived from the following formula:

$$\frac{\partial \rho k}{\partial t} + \frac{\partial}{\partial x_j}\left[\rho(u_j - V_j)k\right] = \widetilde{P}_k - D_k + \frac{\partial}{\partial x_j}\left[(\mu + \sigma_k \mu_t)\frac{\partial k}{\partial x_j}\right] \tag{8}$$

$$\frac{\partial \rho \omega}{\partial t} + \frac{\partial}{\partial x_j}\left[\rho(u_j - V_j)\omega\right] = P_\omega - D_\omega + \frac{\partial}{\partial x_j}\left[(\mu + \sigma_\omega \mu_t)\frac{\partial \omega}{\partial x_j}\right] + CD_{k\omega}(1 - F_1) \tag{9}$$

$$CD_{k\omega} = 2\rho\sigma_{\omega 2}\frac{1}{\omega}\frac{\partial k}{\partial x_j}\frac{\partial \omega}{\partial x_j} \tag{10}$$

Turbulent viscosity is defined as follows:

$$\mu_t = \frac{\rho k a_1}{max\left(a_1\omega, |S_{ij}|F_2\right)} \tag{11}$$

where $\mu_t$ is the turbulent viscosity, and $|S_{ij}| = \sqrt{2S_{ij}S_{ij}}$ is the scalar of the strain rate tensor $S_{ij}$. $F_1$ and $F_2$ are mixing functions that ensure proper selection of the *k-ω* and *k-ε* regions [32]. $P_k$ and $P_\omega$ are production terms; $D_k$ and $D_\omega$ are dissipation terms. Their expressions are referred to in [31] for the sake of brevity.

The cavitation model based on the vapor volume fraction transport equation is used to solve the mass transfer of the vapor–liquid phase change caused by cavitation. The governing equation is expressed as follows:

$$\frac{\partial(\rho_v \alpha_v)}{\partial t} + \nabla(\rho_v \alpha_v u) = \dot{m}_{vap} - \dot{m}_{con} \tag{12}$$

where $\dot{m}_{vap}$ and $\dot{m}_{con}$ are the mass transfer rates of vaporization and condensation, proposed by Zwart et al. [33], and the formulas are as follows:

$$\dot{m}_{vap} = F_{vap}\frac{3\alpha_{nuc}(1 - \alpha_v)\rho_v}{R_b}\sqrt{\frac{2}{3}\frac{max(p_v - p, 0)}{\rho_l}} \tag{13}$$

$$\dot{m}_{con} = F_{con}\frac{3\alpha\rho_v}{R_b}\sqrt{\frac{2}{3}\frac{max(p - p_v)}{\rho_l}} \tag{14}$$

where $F_{vap}$ and $F_{con}$ are the empirical coefficients of vaporization and condensation rates [34], $P_v$ is the vaporization pressure, $R_b$ is the bubble radius, and $\alpha_{nuc}$ is the volume fraction of nucleation point. The reference values of the above parameters are $R_b = 1 \times 10^{-6}$ *m*, $\alpha_{nuc} = 5 \times 10^{-4}$, $F_{vap} = 50$, and $F_{con} = 0.01$ by default. Through the above equations, the fluid modeling can be defined in the CFD software, and the simulation of the fluid transport in the pump model can be carried out by setting the boundary conditions and initial conditions.

### 2.3. Boundary Conditions and Settings

For the fluid boundary conditions, the total pressure of 1.0 atm is selected at the inlet, and the mass flow rate determined by the simulated flow rate is specified at the exit. During the cavitation modeling, the inlet pressure is reduced successively to achieve different cavitating conditions. In addition, as a rotating part, the domain of the impeller is set to the rated speed, and the blades and front and rear cover plates are set as rotating walls. The no-slip condition is defined on all walls. When setting the interfaces between different components, the interfaces between the impeller and the inlet section and between the impeller and the volute are set as the dynamic-static interface while the other interfaces are set as the static interface. Table 3 presents the material properties of three cooling media at 20 and 60 °C used in the simulations [35], wherein the ethylene glycol (EG) aqueous solution and propylene glycol (PG) aqueous solution are prepared with the organic solvent of 60% in volume fraction. EGaq and PGaq are defined to represent the EG aqueous solution and the PG aqueous solution, respectively.

**Table 3.** Material properties of three cooling medium at different temperatures.

| | Temperature/$t$ (°C) | Saturation Pressure/P(bar) | Viscosity/$mPa*s$ | Density/$\rho(kg/m^3)$ | Specific Heat Capacity/ $c(J*kg^{-1} K^{-1})$ |
|---|---|---|---|---|---|
| EGaq | 20 | 1582 | 5.38 | 1086.27 | 3084 |
| | 60 | 13,600 | 1.69 | 1063.69 | 3258 |
| PGaq | 20 | 1760 | 10.04 | 1048.25 | 3339 |
| | 60 | 15,435 | 2.22 | 1020.66 | 3515 |
| Water | 20 | 2330 | 1.005 | 998.2 | 4183 |
| | 60 | 1582 | 5.38 | 1086.27 | 3084 |

The governing equations are discretized by the finite volume method in space. In the steady-state calculation, the iteration steps are set as 1000. The second order discretization schemes are applied for the advection terms and the turbulent parameters. In the unsteady numerical simulation, the time step is defined as the impeller rotates the 2° in each time step. The transient term adopts the second-order backward Euler scheme. The basic settings of cavitation and non-cavitation conditions are the same except for the cavitation model. Firstly, the steady-state numerical simulation is carried out under the condition of non-cavitation. After the calculation converges, the cavitation model is activated. A steady-state simulation of 1000 steps takes 9 h with 8 cores of 2.5 GHz while a transient simulation of 0.054 s (10 rotating cycles) takes 51 h with 16 cores of 2.5GHz. For transient simulation, the number of iterations in each time step is set as 10.

The mesh independence of the simulation result is studied. Five sets of grids with different numbers of control volumes are created for each type of blade, and the dimensionless coefficients of pressure head as defined in Equation (15) are calculated

$$\varphi = \frac{H}{u_2^2 / 2g} \tag{15}$$

where $u_2 = \pi D_2 n / 60 \ m/s$ is the circumferential speed, $D_2$ is the impeller outside diameter, $H$ is the pressure head at the outlet and g the gravitational acceleration. The meshes in the flow channels of the CFD pump model are shown in Figure 2. Hybrid meshes are employed in the mesh of the pump model. Structured meshes are created in the components of impellers while unstructured meshes are formed in the components of volute. Several boundary layer meshes are defined on all wall surfaces. The thickness of the first boundary layer is 0.05 mm, and 10 boundary layers are grown with a growth rate of 1.1. y+ values are larger than 30 and less than 100 on the major wall surfaces. Figure 3 presents the results, and it is found that the pressure head coefficient reaches a saturated value as the fourth mesh is used for both the cylindrical blade impeller and the splitter blade impeller. Therefore, the fourth meshes are chosen for the following simulations.

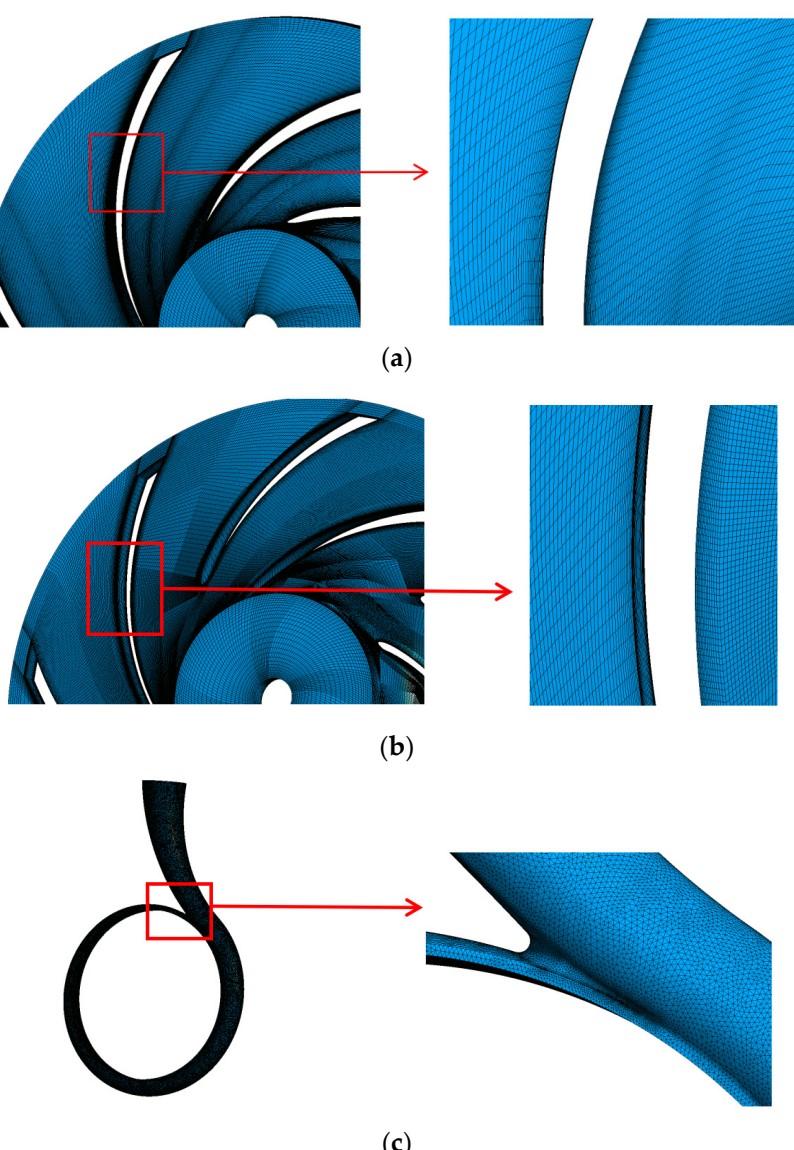

(a)

(b)

(c)

**Figure 2.** Meshes in the flow channels of the CFD pump model: (**a**) cylindrical blades impeller; (**b**) splitter blades; (**c**) volute.

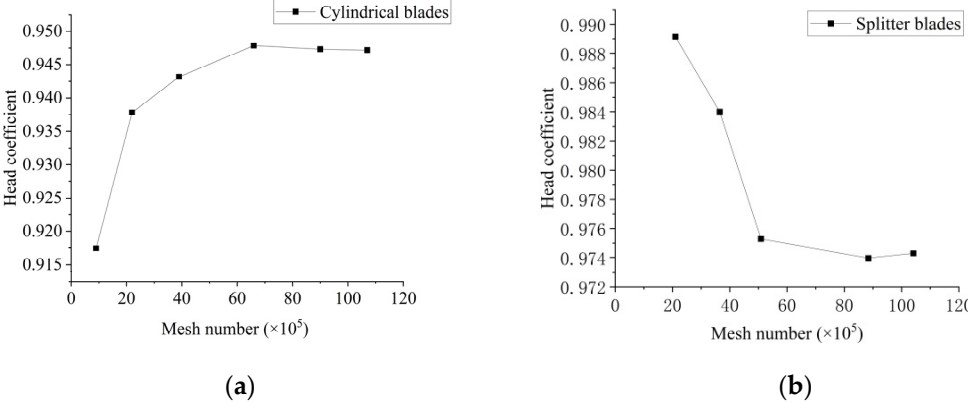

**Figure 3.** Verification of mesh independence: (**a**) cylindrical blades; (**b**) splitter blades.

## 3. Experimental Validation

### 3.1. Experimental Facilities

The authenticity of the numerical model is validated with the experiment performed in a tested pump model. The pump model is made of plexiglass for high-speed photography to study the cavitation characteristics in the pump. Due to the high rotation speed, the experiments are difficult to perform in the prototype pump. Therefore, a speed-reduced model based on the law of similarity of pump is created with the same specific-speed of the prototype pump. The conversions of the design parameters based on the law of similarity are expressed as:

$$\frac{Q_m}{Q} = \left(\frac{n_m}{n}\right)\left(\frac{D_{2m}}{D_2}\right)^3 \tag{16}$$

$$\frac{H_m}{H} = \left(\frac{n_m}{n}\right)^2\left(\frac{D_{2m}}{D_2}\right)^2 \tag{17}$$

$$\frac{P_m}{P} = \left(\frac{n_m}{n}\right)^3\left(\frac{D_{2m}}{D_2}\right)^5 \tag{18}$$

In the experiments, the impellers of both cylindrical and splitter blades and the associated volute are made of plexiglass for visualizing the cavitating flow in the pump model. Figure 4a,b show the impeller models of cylindrical and splitter blades, respectively. Figure 4c shows the on-site pump model. In order to validate the accuracy of the numerical procedures, the physical pump model was tested at 1450 r/min, and the hydraulic and cavitation performances were obtained from the experiments.

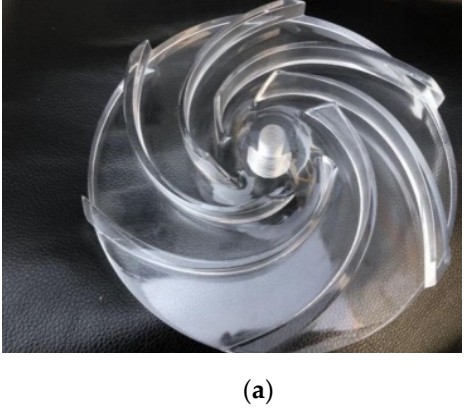

(**a**)

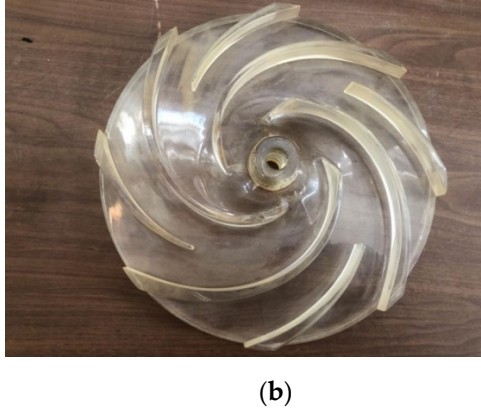

(**b**)

**Figure 4.** *Cont.*

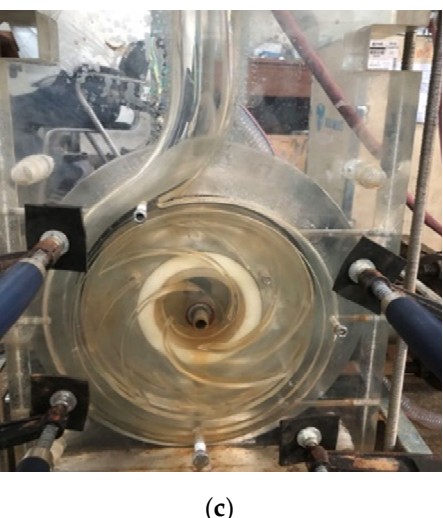

(**c**)

**Figure 4.** (**a**) Impeller mode of cylindrical blades; (**b**) impeller model of splitter blades; (**c**) on-site physical pump model.

A testing loop is built to measure the hydraulic and cavitation performances of the tested pump model and to photograph the cavitation in the pump model. As shown in Figure 5, the loop is mainly composed of a closed water tank with a vacuum pump (for cavitation test), inlet and outlet valves, the pump model, inlet and outlet pipelines, vortex flowmeter, inlet and outlet pressure transmitter, and electrical control cabinet. A high-speed camera as shown in Figure 6 is used to visualize the cavitation inside the pump. The adopted high-speed camera model is i-speed3 high-speed camera with a resolution of $1028 \times 1024$ at a maximum of 2000 frames per second.

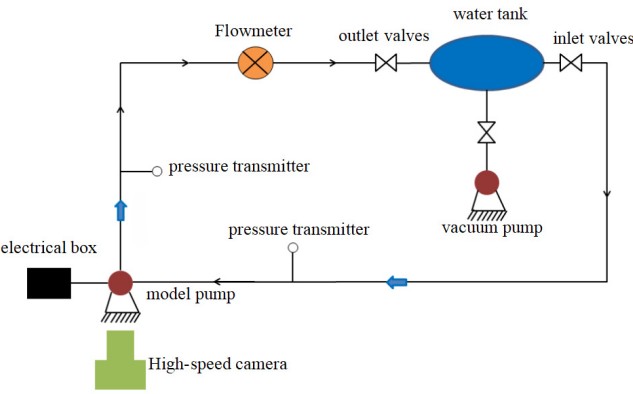

**Figure 5.** Schematic diagram of the testing loop.

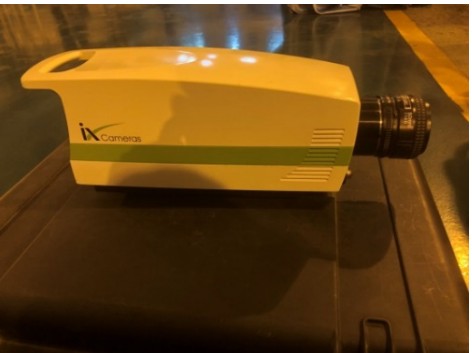

**Figure 6.** High-speed camera.

### 3.2. Validation of Hydraulic and Cavitation Performances

The measured parameters were normalized before presenting in the figures. The dimensionless coefficient of the pressure head is calculated in Equation (15). The dimensionless coefficients of flow rate and cavitation number are expressed as:

$$\psi = \frac{Q}{\pi D_2 b_2 u_2} \tag{19}$$

$$\sigma = \frac{P_{in} - p_v}{1/2\rho u_2^2} \tag{20}$$

where $P_{in}$ is the pressure at inlet; $P_v$ is the saturated pressure of the fluid, and $u_2$ is the circumferential speed of impellers. $b_2$ is the outlet width of the impeller.

In order to compare with the experiment, the numerical model of the tested pump model was built based on the converted design parameters. Both the cylindrical and splitter blades were simulated in the numerical model. The procedure of the numerical simulation is similar to that of creating the numerical model for the prototype except for defining the different geometries and boundary conditions. The information of the numerical procedure will not be repeated here for the sake of brevity.

By taking the simulation result of single-phase flow as the initial value, the cavitating flow in the tested pump was numerically simulated. Figure 7 shows the cavitation curves obtained from both the experiment and simulation. It can be seen that the characteristics of the cavitation curves between the simulation and experiment are in good agreement. The pressure head coefficients remain basically constant at first and then decrease with the decreasing cavitation number. When the cavitation number is large, the pressure heads of the two pumps are almost equal, indicating that the impeller of splitter blades can obtain the same pressure head as the cylindrical blades. However, when the cavitation number is reduced to the cavitating state, the decreasing rate of the pressure head coefficient of the impeller of splitter blades is smaller than that of the impeller of cylindrical blades. This phenomenon appears in both the simulation and experimental results, which demonstrates that the impeller of splitter blades possesses better cavitation performance. It is noted that the predicted pressure heads in the simulation are larger than those in the experiment. This discrepancy is attributed to the simplifications made in the geometry of the numerical model. When severe cavitation occurs under small cavitation numbers, the pressure head coefficients of the simulation are decreased significantly more than those in the experiment. From the photographs of the cavitating flow in the tested pump presented in Section 3.3, it is found that the numerical model over-predicts the cavitation zone and the resulting drop in the pressure head.

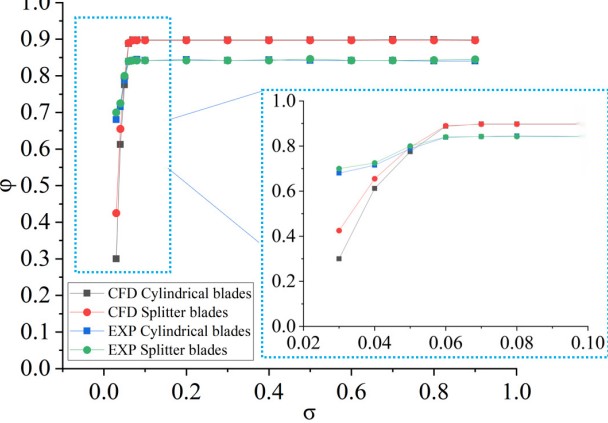

**Figure 7.** Cavitation curves of the pressure head coefficients of two impellers from simulation and experiment.

### 3.3. Validation of Cavitation Visualization

The cavitation zones in the impellers with two types of blades are shown in Figures 8 and 9 under the inlet pressures of 0.09 atm (σ = 0.058) and 0.07 atm (σ = 0.04), where different degrees of cavitating flows are observed in the photographs. The cavitation zone is represented by the iso-surface of vapor volume fraction $\alpha_v$ = 0.5 in the simulation results. It is seen that cavitation occurs slightly in the flow channels of both impellers at σ = 0.058. The original area for cavitation in both impellers is located at the suction side of blades near the inlet. Compared with the cylindrical blades, the cavitations in the splitter blades are distributed more uniformly and the cavitation intensity is smaller in each flow passage.

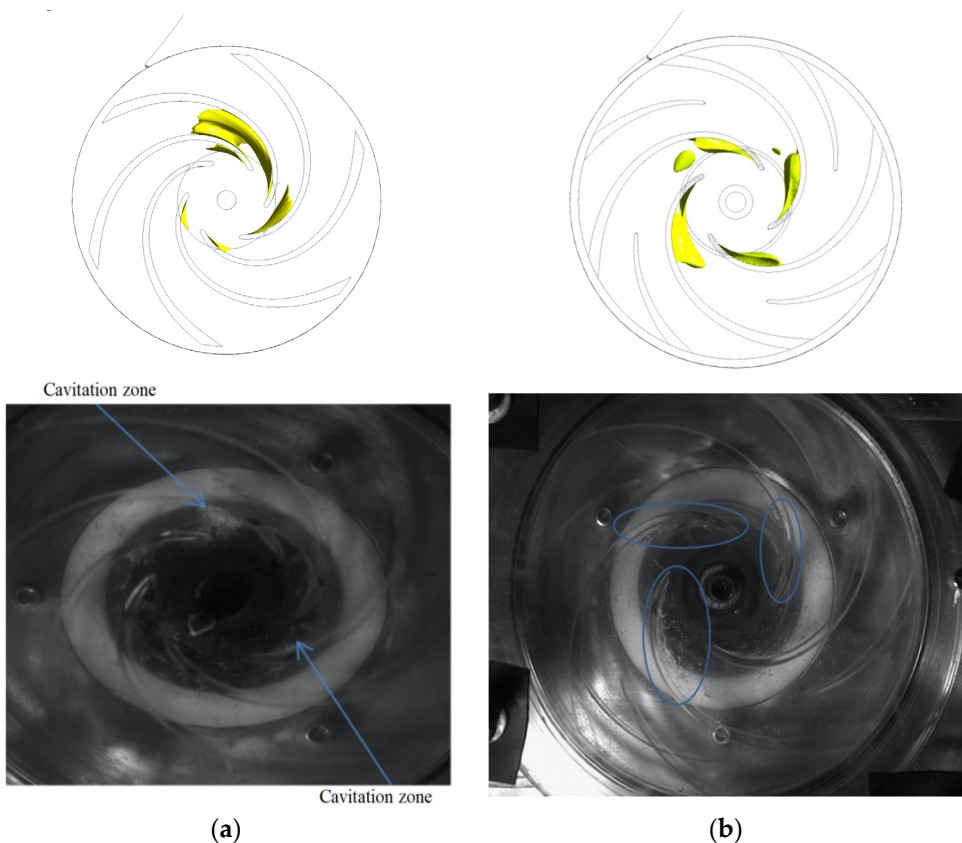

**Figure 8.** Cavitation zones in both cylindrical and splitter blades at σ = 0.058. (**a**) Cylindrical blades; (**b**) Splitter blades.

When σ = 0.04, severe cavitation has occurred in both impellers, and the pressure head has also decreased greatly. However, in the splitter blades, the cavitation degree is significantly less than that in the cylindrical blades as shown in both the simulations and photographs, and the same feature is also applied to the pressure head coefficient. The cavitation performance of the impeller of splitter blades is better than that of the cylindrical blades. It is also noted that the depicted cavitation zones in the simulations are smooth and distinct from the water body while the cavitation bulbs in the high-speed photographs are fragmentized and mixed with the water body. This is attributed to the employed turbulence model in which the Reynolds-averaged velocity field is solved and the detailed turbulence field is missed. However, the simulation can predict the hydraulic and cavitation performances of the pump model with reasonable accuracy. Based on the above validations, the same procedure of the numerical simulation will be applied in the following simulations for the prototype.

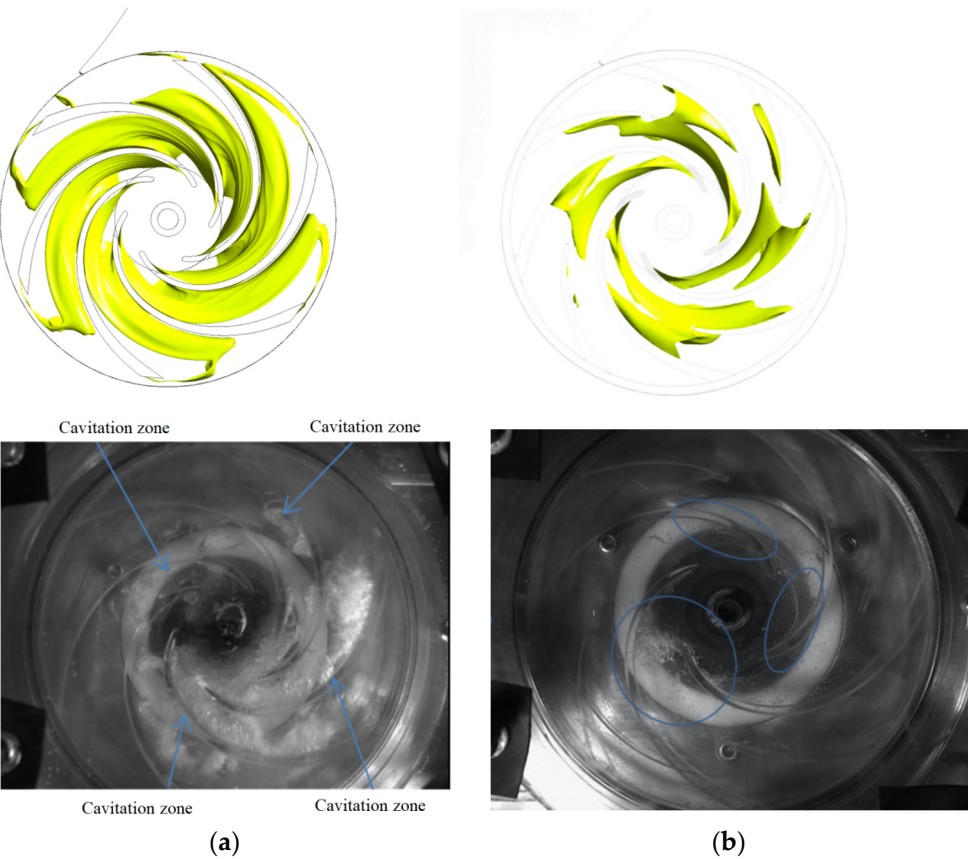

**Figure 9.** Cavitation zones in both cylindrical and splitter blades at σ = 0.04. (**a**) Cylindrical blades; (**b**) Splitter blades.

## 4. Results and Discussion

In the preceding section, the authenticity of the numerical model was validated with the experiment in terms of both the hydraulic and cavitation performances. In this section, the numerical model of the prototype pump is employed to study its performance at high rotation speed. Based on the cylindrical and splitter impellers, the effects of coolant and working temperature on the characteristics of the hydraulic performance and cavitation are investigated in the pump of high rotation speed.

### 4.1. Characteristics of Pump Hydraulic Performance

The pump hydraulic performance was numerically simulated for conveying water, EGaq, and PGaq at different flow rates. The pressure head and pump efficiency of the pumps with cylindrical and splitter blades are shown in Figures 10 and 11, respectively. It is shown that the pressure head and pump efficiency for conveying water are higher than those for conveying EGaq and PGaq because the water viscosity is much smaller, resulting in less energy dissipation. A comparison between the two solutions shows that the performance of EGaq is slightly better than that of PGaq. The pressure head for conveying water reaches 132 m, and its efficiency is 62.9% at the nominal flow rate, which agrees well with the experimental date tested for the prototype. The relatively low efficiency is mainly caused by the energy dissipation of fluid friction occurring in the gap between the impeller of high rotation speed and the static back case. Due to the larger viscosity of the organic solutions, the pressure heads are only in 120~125 m, and the pump efficiencies are in 53~57% for conveying EGaq and PGaq. The point of the maximum pump efficiency agrees well with the nominal flow rate, indicating a good design for the hydraulic performance. It is also found the two organic solutions have a wider range of high efficiency, compared with that possessed by water. The type of impeller blade has little influence on pump performance, since the characteristics of the performance curves are very similar between

the two impellers. Compared to the impeller of cylindrical blades, the impeller of splitter blades shows a slightly better performance for conveying EGaq than PGaq in terms of the predicted pressure head and efficiency.

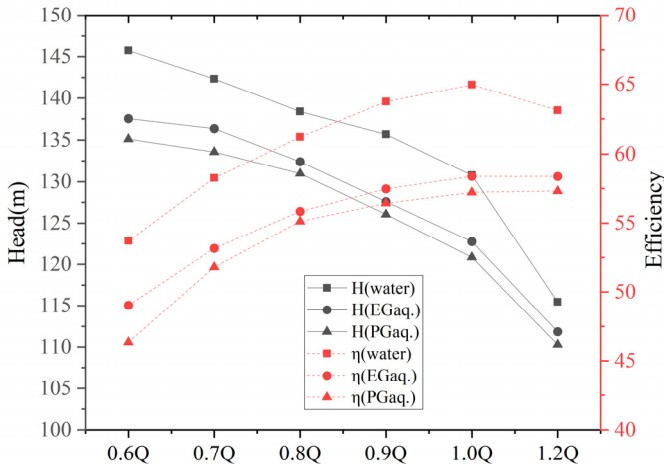

**Figure 10.** Characteristic curves of the pump with cylindrical blades.

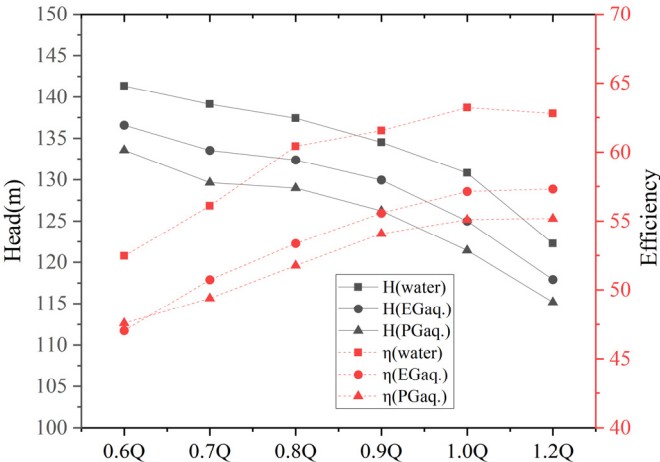

**Figure 11.** Characteristic curves of the pump with splitter blades.

### 4.2. Characteristics of Pump Internal Flow

The characteristics of pump internal flow for conveying water, EGaq, and PGaq at three typical flow rates are investigated. The streamlines and contours of fluid velocity in the impellers of cylindrical and splitter blades are shown in Figures 12 and 13. In general, fewer vortices and reserve flow are observed in the flow channels at the low, nominal, and high flow rates, indicating the designed impellers have good control for the internal flow. Specifically, the streamlines in the impeller of cylindrical blades for conveying EGaq and PGaq are smoother than those in water at $0.7Q_d$ while the streamlines at $1.0Q_d$ and $1.2Q_d$ are very similar among three media. The streamlines in the impeller of splitter blades are less smooth than those in the impeller of cylindrical blades. Small vortex occurs in water near the outlet of the impeller at $0.7Q_d$. EGaq and PGaq show better internal flow fields than water, and the internal flow in the impeller of splitter blades could be further improved by optimizing the shape and position of the short blades.

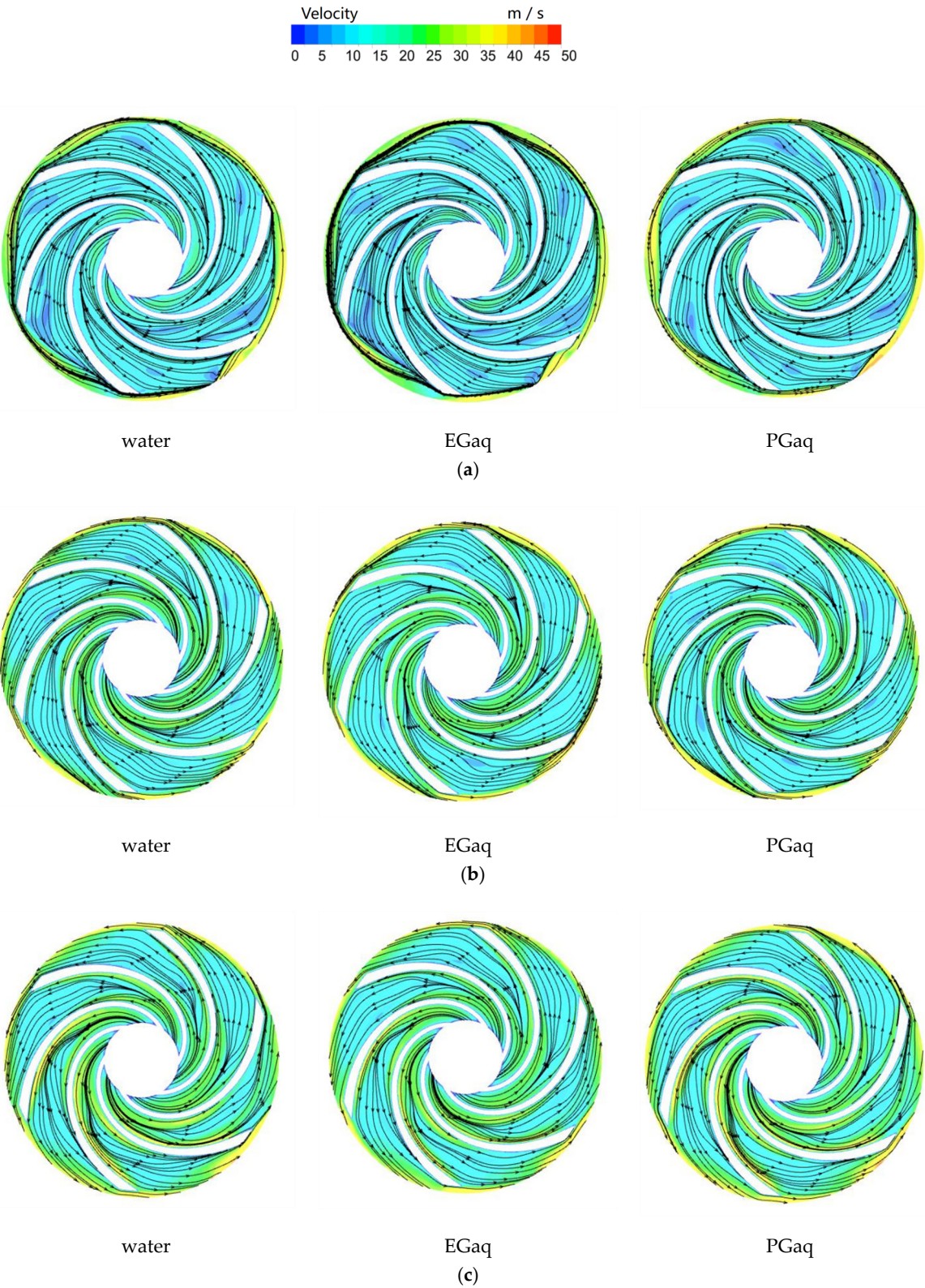

**Figure 12.** The velocity of the middle plane of the cylindrical blade impeller at different flow rates. (**a**) $0.7Q_d$; (**b**) $0.7Q_d$; (**c**) $0.7Q_d$.

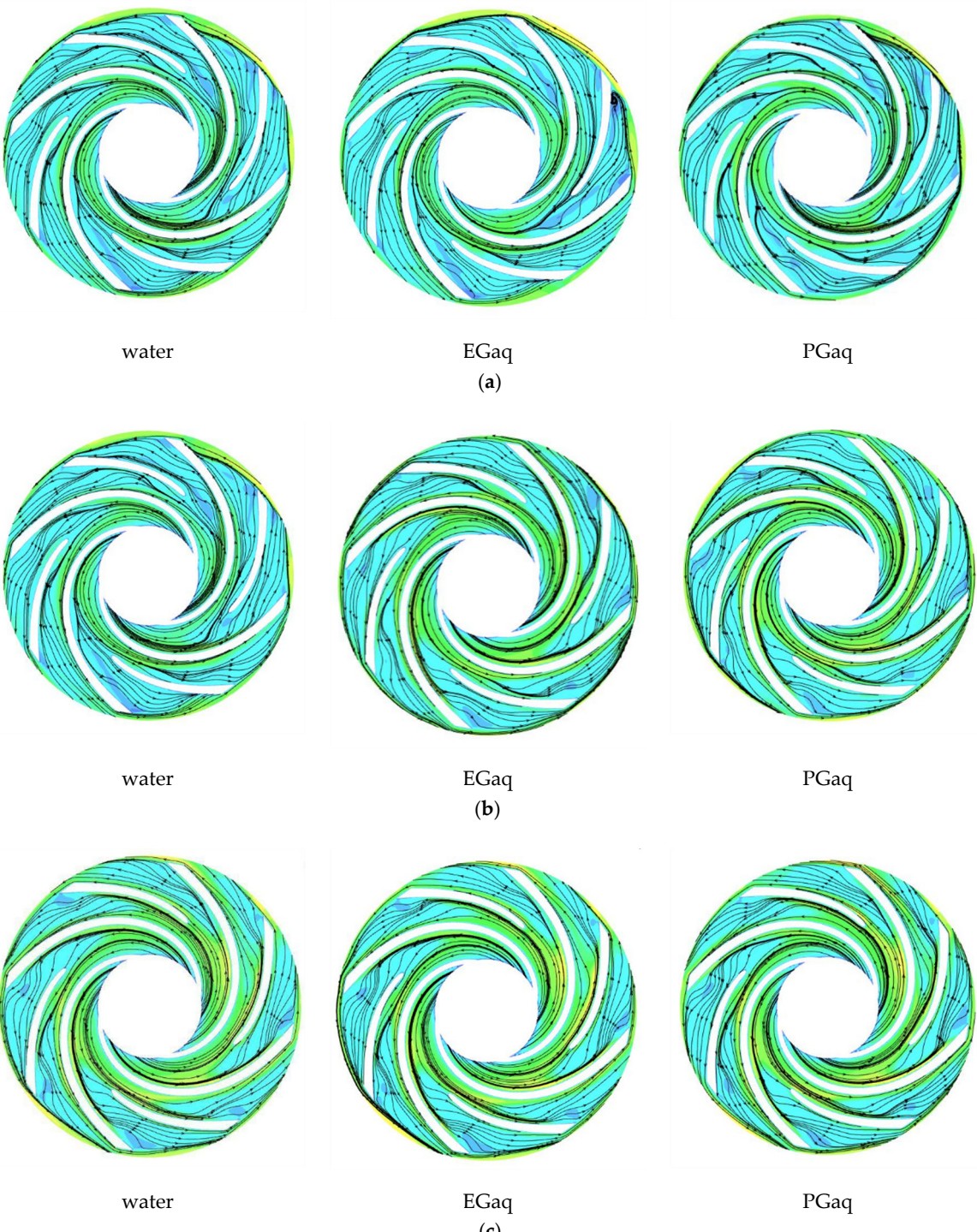

**Figure 13.** The velocity of the middle plane of the splitter blade impeller at different flow rates.
(**a**) $0.7Q_d$; (**b**) $0.7Q_d$; (**c**) $0.7Q_d$.

### 4.3. Characteristics of Pump Cavitation

The characteristics of pump cavitation for conveying water, EGaq, and PGaq are numerically investigated at different working temperatures, i.e., 20 and 60 °C. The correlations of cavitation number and inlet pressure for the impellers of cylindrical and splitter blades are presented in Figures 14–17. At 20 °C, the pressure head of the pump with cylindrical impeller blades as shown in Figure 14 can be retained until the inlet pressure is decreased to 0.5 atm. The pressure heads for conveying water, EGaq and PGaq are slightly decreased

at 0.4 atm, indicating an inception of cavitation in the pump. The pressure heads are significantly decreased by 8.8%, 8.1%, and 9.5% at 0.3 atm, indicating severe cavitations in the pump. When the working temperature is at 60 °C, the pressure head as shown in Figure 15 can be retained until the inlet pressure is decreased to 0.6 atm. The pressure heads begin dropping at 0.5 atm, and they are decreased by 13.6%, 10.9%, and 11.8% at 0.4 atm, respectively, for conveying water, EGaq, and PGaq. The pressure head of the pump with splitter blades as shown in Figure 16 can be retained also at 0.5 atm, which is coincident with the pump with cylindrical blades at 20 °C. The pressure heads for conveying water, EGaq, and PGaq are decreased by 7.2%, 7.2%, and 7.3% at 0.3 atm, respectively. At 60 °C, the pressure heads begin dropping at 0.5 atm, and they are significantly decreased by 16.5%, 9.5%, and 10.1% at 0.4 atm, respectively.

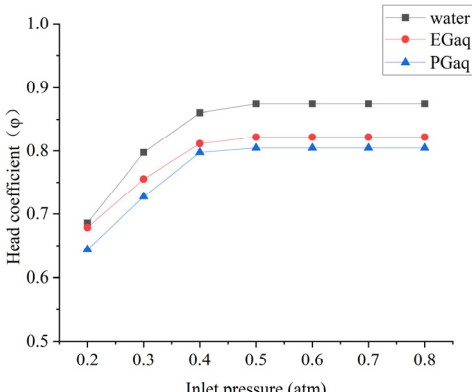

**Figure 14.** Cavitation curve of the pump with cylindrical blades at 20 °C.

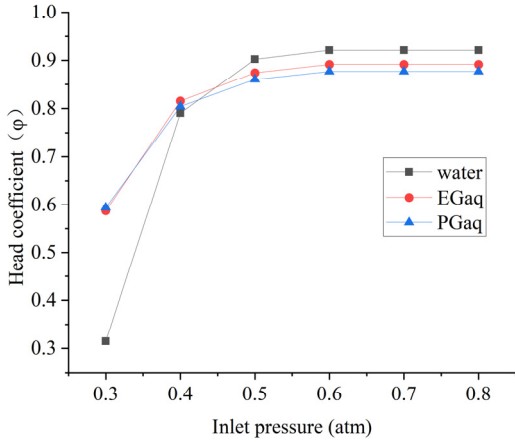

**Figure 15.** Cavitation curve of the pump with cylindrical blades at 60 °C.

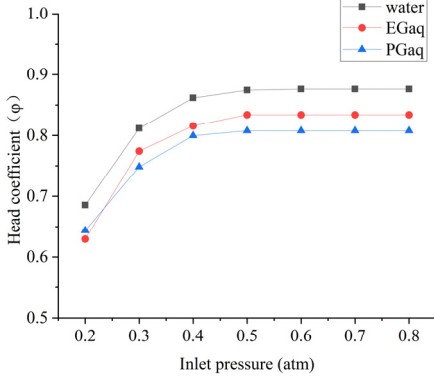

**Figure 16.** Cavitation curve of the pump with splitter blades at 20 °C.

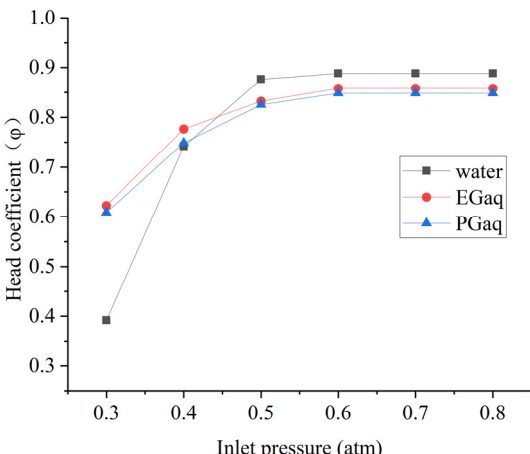

**Figure 17.** Cavitation curve of the pump with splitter blades at 60 °C.

It is found that the inception of cavitation occurs almost at the same inlet pressure between two types of impeller blades. The drops of pressure head in the pump with splitter blades are slightly smaller than those in the pump with cylindrical blades, indicating that the former impeller can improve the cavitation performance of the pump. The higher the working temperature is, the larger the critical inlet pressure for occurring cavitation in the pump is. At 60 °C, cavitation occurs at the pump inlet pressure below 0.6 atm, which can be easily encountered in aeronautic environment. Among different cooling media, the descending rate of the pressure head in water when occurring cavitation is much faster than that of the other two organic coolants, especially for the high working temperature. Compared between the two organic coolants, the pressure head in EGaq drops more slowly than that in PGaq under all simulated conditions, which demonstrates that the cavitation performance of EGaq is better than that of PGaq.

According to the above analyses, the cavitation zones in the different pump impellers are represented by depicting the iso-surface of the vapor volume fraction of 0.5 as shown in Figures 18 and 19. The cavitation zones are shown at 20 °C with the inlet pressure of 0.3 atm and 60 °C with the inlet pressure of 0.4 atm. In Figure 18, it is shown that the volumes of cavitation zones in the three coolants are very close at the low working temperature. They occur firstly at the suction side of each blade. However, the volume of the cavitation zone in water is significantly increased at the high working temperature while the increase is very limited in EGaq and PGaq. EGaq shows the smallest inflation in the cavitation volume, which agrees well with the results obtained from the analyses on the cavitation curves. The result indicates that the volume of cavitation is related not only to the saturated pressure but also to the viscosity of cooling medium. As the temperature rises, the influence of saturated pressure on cavitation becomes more significant. In the impeller of splitter blades, the cavitation only occurs at the suction side of the long blades, which are extend to the pump inlet. The length of the cavitation zone is much longer than the counterparts in the cylindrical blades. The correlations between the working temperature and the cavitation zone among different coolants are very similar with those in the cylindrical blades. EGaq shows the smallest inflation in the cavitation volume with the increasing temperature.

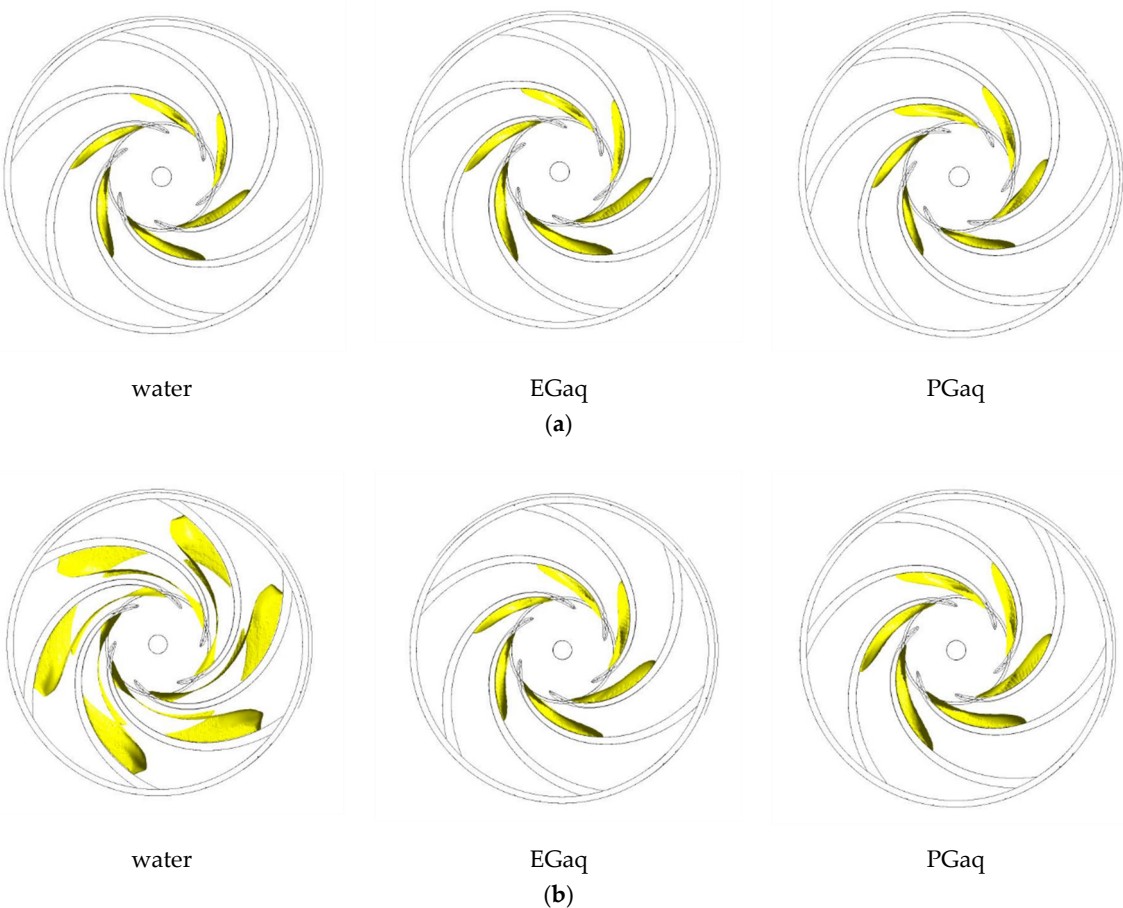

water       EGaq       PGaq

(**a**)

water       EGaq       PGaq

(**b**)

**Figure 18.** Iso-surface of cavitation distribution in the impeller of cylindrical blades. (**a**) 20 °C at 0.3 atm; (**b**) 60 °C at 0.3 atm.

In order to quantitatively estimate the cavitation performances in different conditions, the volumes of the cavitation zones are statistically presented in Figures 20 and 21. It is found that the cavitation performance of water is far less than those of EGaq and PGaq under high working temperature while the cavitation volume of EG is slightly smaller than that of PG. The increasing temperature has less effect on EGaq and PGaq. Between two types of impellers, the cavitation volume in the impeller of splitter blades is slightly smaller than that in the cylindrical blades (only one exception). It is found that the splitter blades can improve the cavitation in the experiments where the pump model works at a low rotation speed (1450 rpm). The finding is supported by the results shown in Figures 7 and 9. However, the cavitation in the pump model working at high rotation speed (11,000 rpm) behaves very differently. The splitter blades at such condition are not helpful in reducing the cavitation in the pump model compared with the cylindrical blades. This may be attributed to the factor that the circumferential speed of the blade is very high at the high rotation speed. Although the splitter blades reduce the number of blades at the inlet and increase the area of the flow channel, the positive effect on reducing the occurrence of cavitation is diminished by the high circumferential speed of the inlet blades. Therefore, it can be concluded that the effect of splitter blades on reducing the cavitation is prominent at a low rotation speed while this effect is marginal at a high rotation speed, such as 11,000 rpm.

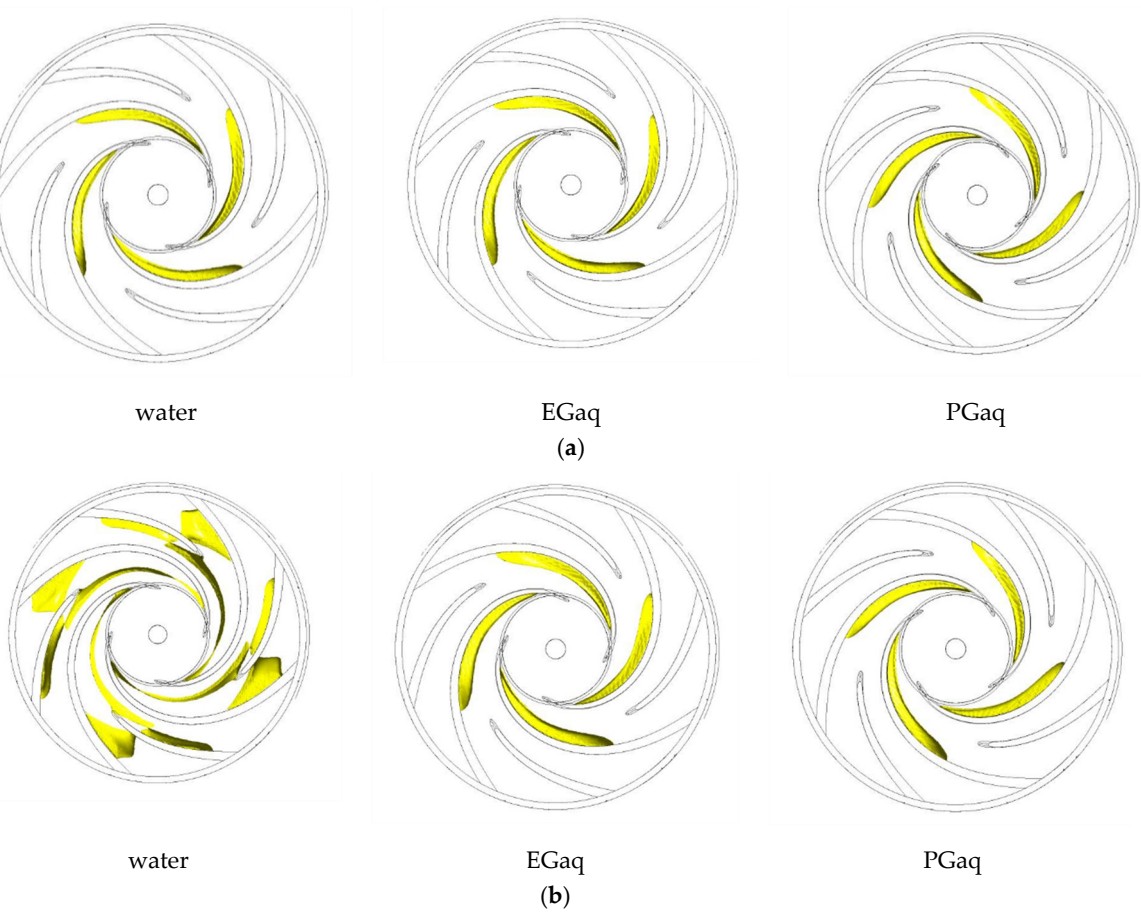

**Figure 19.** Iso-surface of cavitation distribution in the impeller of splitter blades. (**a**) 20 °C at 0.3 atm; (**b**) 60 °C at 0.3 atm.

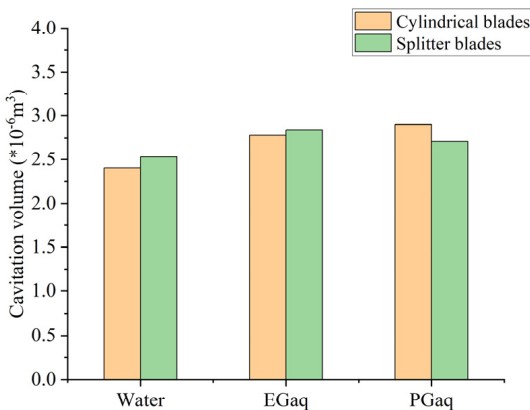

**Figure 20.** Volumes of cavitation zone in different impellers and coolants at 20 °C with an inlet pressure of 0.3 atm.

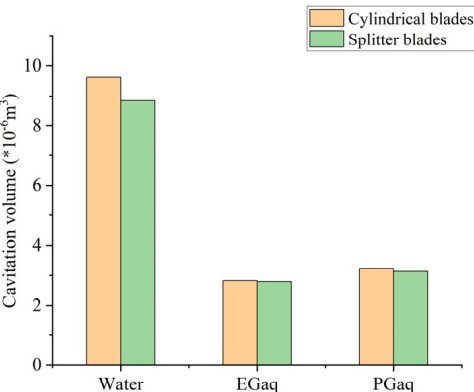

**Figure 21.** Volumes of cavitation zone in different impellers and coolants at 60 °C with an inlet pressure of 0.4 atm.

## 5. Conclusions

In this paper, a pump model with two types of impellers was designed based on the design parameters of an aeronautic cooling pump. A design method consisting in enlarging the design flow rate was used for designing the pump model, and the corresponding CFD model was built and simulated. Moreover, the accuracy of the numerical procedures was verified by comparing the simulation results with the performed experiments in terms of both hydraulic and cavitation performances.

Based on the verified numerical model, the effects of different types of impellers, cooling media, and working temperature on the hydraulic and cavitation performances of the aeronautic cooling pump were numerically investigated. It is found that the pressure head and pump efficiency for conveying water are higher than those for conveying EGaq and PGaq at 20 °C while the hydraulic performance of EGaq is slightly better than that of PG. In addition, two organic solutions have a wider range of high efficiency, compared with water. The impeller of splitter blades shows a slightly better performance for conveying EGaq than PGaq in terms of the predicted pressure head and pump efficiency. As for the characteristics of pump internal flow, the streamlines in the impeller of splitter blades are less smooth than those in the impeller of cylinder blades at low, nominal, and high flow rates. EGaq and PGaq show better internal flow fields than water, and the internal flow in the impeller of splitter blades could be further improved by optimizing the shape and position of the short blades. Comparing EGaq and PGaq, there is no significant difference in the internal flow field.

It is found that the cavitation in the pump is very similar at 20 °C for the three coolants. However, water has been severely cavitated at 60 °C with an inlet pressure of 0.4 atm. The cavitation performance of water is far less than that of EGaq and PGaq under high working temperature. The increasing temperature has less effect on EGaq and PGaq. The cavitation volume of EGaq is smaller than that of PGaq, and the cavitation volume in the splitter blades is slightly smaller than that in the cylindrical impeller. Since the coolant is usually working at high temperature, EGaq and PGaq are much better than water as the coolant. It is suggested to use EGaq as the first option. At 60 °C, which most often is the working condition of the cooling pump, the cavitation volume in the splitter blades is slightly less than that in the cylindrical blades in either water or any of the two aqueous solutions. However, the improvement by using the splitter blades is very limited. The splitter blades can improve the cavitation performance, but at the same time, they diminish the efficient performance of the pump. Therefore, the design of the short blades should be optimized in order to obtain a smooth internal flow field.

**Author Contributions:** Conceptualization, A.W. and R.Z.; formal analysis, A.W. and R.Z.; data curation, A.W.; writing—original draft preparation, A.W. and R.Z.; writing—review and editing, A.W., R.Z., F.W, D.Z. and X.W.; funding acquisition, F.W., D.Z. and X.W. All authors have read and agreed to the published version of the manuscript.

**Funding:** The authors are grateful for the financial supports from the National Natural Science Foundation of China (Grant No.: 52176038), Key R & D projects in Jiangsu Province (Grant No.: BE2021073), Aeronautical Science Fund (No. 201728R3001) and Primary Research & Development Plan of Shandong Province (No.: 2019TSLH0304).

**Institutional Review Board Statement:** Not applicable.

**Informed Consent Statement:** Not applicable.

**Data Availability Statement:** The data used to support the findings of this study are included within the article.

**Conflicts of Interest:** The authors declare no conflict of interest.

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
