# Peer review of "Effects of Coolant and Working Temperature on the Cavitation in an Aeronautic Cooling Pump with High Rotation Speed"

_machines, doi:10.3390/machines10100904_

Round 1

Reviewer 1 Report

Dear authors,

Thank you for the great effort studying and presenting the cavitation phenomena and its drastic effects in the pump's performance. I have some comments in the attached pdf, and I am going to list them here too:

** In equation (2), the left hand side (ns) needs to be (ns')

** In lines 129-130: Are K1 and K2 empirically determined? and are constant for all pumps designs?

** In line 151: Could you please confirm if you used Volume of Fluid (VOF) or another multiphase model?

** In line 201: Could you please clarify why you chose such time step?

In this kind of cavitating turbomachinery, the time scales will depend on either rotation, flow velocity in pipes, or the speed of cavitation phenomena.

Also, how this chosen time step affect the Courant number?

** In line 207: Two questions:

1- Why is the simulation not carried out on the same number of cores? 

Sometimes, parallel computing on different number of cores adds another source of error.

2- How many inner iterations in each time step?

** In line 231: There is a typo (poexiglass) to (plexiglass)

** In lines 304-306: This part needs more clarification. I think the discrepancy comes from different reasons including turbulence, but the major factor might be the multiphase model which seems to be the volume of fluid approach. Please discuss this part more and how the VOF might have some limitations in representing the vapor bubbles. 

** Are figures 13 and 14 based on the data of the last time step or time averaging?

** In lines 413-414: 

- Could you please discuss the viscosity differences between the three liquids?

- Is there a (negative/positive/or no) correlation between viscosity and cavitation? 

** In line 415: This statement requires some elaboration: why the saturation pressure becomes more significant?

** In line 422-430: I recommend this paragraph to be below figure 19 for easier linking with the corresponding figures (Fig. 20 and 21).

** In line 424: The word "performance" could mislead the readers as it is a metric of functionality, while cavitation is adverse phenomenon. I recommend editing the comparison statements with simpler factual ones like (cavitation volume in water is more than EGaq. and PGaq. at high temperature). 

** In Figures 20 and 21:

Two points aside of the water data:

- Figure 20 shows contradicting behavior between the cylindrical and splitter blades. One type is generating more cavitation than the other blade with each liquid. Does this give an upper hand to the splitter type?

- In Figure 21, the temperature did not affect the cavitation content at any of the aqueous liquids. Also, the cavitation content is almost equal between the two blade designs. 

- These cavitation volume figures make it less confirmatory that the splitter design is better than cylindrical. Or the even the aqueous liquids are better than water at low temperature.

I hope my comments/recommendations help in a better review.

Best of luck

Reviewer 2 Report

Dear Authors

I find your paper very interesting. I have found some small inaccuracies in the paper. Please, see your originally submitted file (if you have not received it, please contact Editor). When you answer all questions and correct shortcomings, please write explanations in text as much and as clear as possible.

Fig.1. The presented domain and text do not contain any information about the sizes of a pump. It should be corrected.

General. No information is presented about the mesh, dimensionless distance Y+, etc. It also should be corrected.

General. No information about solver is also provided.

Table 3. The results presented in the table would be more impressive if they were presented in a figure.

L256. There is missing radius at which the circumferential speed is calculated.

Fig.7. Use different colours for all 4 cases. Now the differences are not readable for the red and black lines.

Fig.14 and 15. Consider to overlap both figures. It would be more impressive. Use different colours.

Fig.16 and 17. As above.

Conclusion. “The splitter blades can improve the cavitation performance…” In my opinion it should be highlighted that splitter ‘can improve the cavitation performance’ but at the same time it generates less efficient performance of the pump.

Reviewer 3 Report

The design idea with splitter is interesting. The paper clearly describes the validation of the CFD model and results of the numerical calculation. Conclusion are supported by results. The paper in general is written very well. There are very minor English mistakes that can be easily fixed, for example in the abstract (line 19 .. "to represented" should be "to represent").

Also, please change the symbol for any mass rate with m with dot on top of it as this is a very commonly used symbol.

And, please simply use EGaq and PGaq without dots at the end for the names of the fluids used in the experiment. This will remove confusion when the names are at the end of sentence and need to be followed by a period.

Also, please add pictures of blades with cavitations. This would make the validation even stronger. 

The citations are now sufficient. But discussion on cavitation has been so long and there are large number of literatures on this subject. So, please add about five more papers.

Lastly, on the captions - please point out key important messages you want to tell readers using the figures. I think this will make easy for readers to read.
